# House Dust Mite Subcutaneous Immunotherapy and Lung Function Trajectory in Children and Adolescents with Asthma

**DOI:** 10.3390/children9040487

**Published:** 2022-04-01

**Authors:** Kazutaka Nogami, Mizuho Nagao, Takafumi Takase, Yasuaki Yasuda, Shingo Yamada, Mayumi Matsunaga, Miyuki Hoshi, Kana Hamada, Yu Kuwabara, Takeshi Tsugawa, Takao Fujisawa

**Affiliations:** 1Department of Pediatrics, Sapporo Medical University School of Medicine, Sapporo-shi 060-8543, Japan; knogami@sapmed.ac.jp (K.N.); tsugawat@sapmed.ac.jp (T.T.); 2Allergy Center and Department of Clinical Research, Mie National Hospital, Tsu 514-0125, Japan; nagao.mizuho.yt@mail.hosp.go.jp (M.N.); takatak817@gmail.com (T.T.); milaninter20000@yahoo.co.jp (Y.Y.); shingoru529@yahoo.co.jp (S.Y.); kirsche1129@yahoo.co.jp (M.M.); deep_snow_star@yahoo.co.jp (M.H.); kana_ham8@yahoo.co.jp (K.H.); yu.kuwabara2@gmail.com (Y.K.); 3Department of Pediatrics, Ehime University Graduate School of Medicine, Toon 791-0295, Japan

**Keywords:** immunologic, asthma, spirometry, immunoglobulin E, eosinophils

## Abstract

Background: Allergen-specific immunotherapy is currently the only disease-modifying treatment for allergic asthma, and it has been shown to improve control of asthma while reducing both drug use and asthma exacerbations. However, its effects on lung function—especially its long-term effects—remain controversial. We aimed to identify factors associated with a possible beneficial effect of allergen-specific immunotherapy on lung function in asthma by retrospectively evaluating the long-term changes in lung function in children with asthma who received house dust mite subcutaneous immunotherapy (HDM-SCIT). Methods: We enrolled children with asthma who had undergone HDM-SCIT for more than 1 year. Clinical information and lung function measurements were retrieved from the electronic chart system. To characterize the trajectory of lung function change, we performed linear regression analysis to evaluate the maximal expiratory flow at 50% of the forced vital capacity during two periods: before and during HDM-SCIT. Slopes from a least-squares regression line for the two periods, i.e., S1 before HDM-SCIT and S2 during HDM-SCIT, were compared. The subjects were then classified into two groups: an improving group (Group I) defined as S2 − S1 > 0, and a declining group (Group D) defined as S2 − S1 < 0. The clinical factors at the start of HDM-SCIT were compared between the two groups. Results: A total of 16 patients were analyzed. Eight patients were classified into each of Group I and Group D. The mean ages were 10.5 and 11.8 years, and the mean treatment periods were 4.1 and 3.9 years. Group I had a significantly lower blood eosinophil count and a significantly higher HDM-specific IgE level than Group D. Logistic regression showed a strong relationship between those two markers and the lung function trajectory. Conclusion: Control of the blood eosinophil count in highly HDM-sensitized patients may increase the beneficial effect of HDM-SCIT on lung function.

## 1. Introduction

Asthma is a heterogeneous inflammatory disease characterized by bronchial hyperreactivity and airflow obstruction. In allergic asthma, respiratory symptoms and underlining airway inflammation are triggered by allergen exposure. House dust mite (HDM) is the most common and important allergen [1], especially in children and adolescents [2]. Allergen-specific immunotherapy (AIT) aims to induce allergen-specific immune tolerance through induction of suppressor cytokines such as IL-10 and transforming growth factor-β that lead to limitation of inflammatory cascade and unresponsiveness to the allergen [3], and AIT is currently the only disease-modifying treatment not only for allergic asthma but also for allergic rhinitis and venom allergy, as well as food allergy. Cumulative evidence indicates that AIT, particularly HDM AIT [4], improves asthma control and the quality of life (QOL), while also reducing exacerbations and long-term drug use [5,6]. Recent evidence-based updates to the US asthma management guidelines recommended subcutaneous immunotherapy (SCIT) as an adjunct therapy to standard pharmacotherapy for mild to moderate persistent asthma [7]. Adverse events are mainly local, and systemic allergic reactions and anaphylaxis is rare [5], which can be managed at well-equipped institutions.

The efficacy of AIT for lung function in asthma, however, is yet to be established [4,8]. In a recent randomized open-label trial, the forced expiratory volume 1 s (FEV1) in the AIT group increased from the baseline, although without statistical significance, whereas it showed no improvement in the standard pharmacotherapy group [9]. Likewise, no consistent results have been obtained with regard to lung function. Although lung function is not a critical outcome of AIT [4], it is still important, particularly in the long-term because a large reduction in lung function would directly affect QOL, as seen in COPD.

Recent cohort studies demonstrated that low childhood lung function or low acquired lung function in early adulthood has lifelong consequences, including the risk of developing COPD [10,11,12,13]. Despite such a dismal prospect, no therapeutic interventions have been proven to prevent lung function decline. In some studies, early intervention with inhaled corticosteroids (ICS) for mild asthma reduced the loss of lung function [14], but recent long-term observations failed to show a preventive effect of ICS on lung function decline [15,16,17]. Newly developed biologics have shown potential [18,19], but more observations are needed to establish their effectiveness. In this context, it is of value to investigate AIT for a possible beneficial effect on lung function in asthma, particularly in early-stage disease.

Here, we investigated long-term changes in lung function in children and adolescents with asthma who underwent HDM-SCIT. Since AIT has not shown a uniformly beneficial effect on lung function, we hypothesized the existence of a subtype of patients who benefit from AIT or factors associated with a favorable outcome. To test that hypothesis, we compared the baseline factors between patients whose long-term lung function improved during HDM-SCIT and those whose function did not improve.

## 2. Materials and Methods

### 2.1. The Study Population

We enrolled children and adolescents <18 years of age with HDM-allergic asthma who underwent HDM-SCIT for ≥1 year at Mie National Hospital. Allergy to HDM was confirmed on the basis of a positive test for specific IgE (sIgE) to HDM (*Dermatophagoides pteronyssinus*) and a history of HDM-related symptoms such as a cough or wheezing upon exposure to a “dusty environment”. The patients were eligible if they had valid spirometry data at 5 or more time-points during the pre-SCIT period of at least 6 months so that changes in lung function could be compared before and during the treatment. SCIT was performed using a standardized HDM extract (ALK-Abellò; Round Rock, TX, USA) as rush immunotherapy followed by a monthly maintenance protocol, as reported elsewhere [20]. During the SCIT maintenance period, spirometry was performed at each injection visit.

Various demographic data, spirometry results, and laboratory test results, including the eosinophil count, total IgE, and allergen-specific IgE, were retrieved from the electronic chart system.

### 2.2. Lung Function

Lung function was measured with a flow-volume spirometer (Chest M.I.; Tokyo, Japan), and predicted values of the parameters were obtained using equations based on the height and age of Japanese children [21]. Since SCIT should be performed when asthma is under control, all the measurements were performed when the patients were asymptomatic and stable. Valid data measured according to the ERS/ATS technical statement [22] were adopted. The maximum expiratory flow at 50% of forced vital capacity (MEF50) was used to evaluate the change, or trajectory, in lung function, as recommended in the European guidelines [4]. The changes in MEF50% predicted before and during HDM-SCIT were estimated for each subject by fitting a least-squares regression line. The slopes were calculated from the fitted line with an equation of y = α + βx, in which x is the number of days from the start of treatment and is designated as S1 for the pre-SCIT period and S2 for the SCIT period. We divided the subjects into 2 groups based on comparison of the S1 and S2 values: patients were classified as Group I (improving group), a favorable trajectory, if S2 − S1 > 0, or as Group D (declining group), an unfavorable trajectory, if S2 − S1 < 0. Changes in percentages of predicted forced expiratory volume in 1 s (FEV1%) and FEV1/forced vital capacity (FEV1/FVC) were also evaluated using the same method.

### 2.3. Statistical Analysis

The baseline clinical data at the start of SCIT were compared using the chi-squared test for categorical variables and the Mann–Whitney U test for continuous variables. A logistic model for predicting a favorable lung function trajectory (Group I) was constructed using variables that were significantly different by univariate analyses. We report the adjusted probabilities of the outcome, the odds ratio (OR), and the 95% confidence interval (CI) based on the model. Goodness-of-fit was tested using the Hosmer–Lemeshow test, and the area under the receiver operating characteristics (ROC) curve for predicting a favorable trajectory was obtained. Wilcoxon’s rank-sum test was used to compare changes in the ICS dose during SCIT. Statistical analyses were performed with GraphPad Prism 9 (GraphPad Software; La Jolla, CA, USA).

## 3. Results

### 3.1. Subjects

A total of 16 patients were eligible for the study, and we analyzed 1158 spirometric measurements after excluding 7 outliers out of the total 1165 by the ROUT method with a false discovery rate <1% [23]. Table 1 summarizes the annual change in MEF50% predicted before and during SCIT in all patients. In accordance with our definition, 8 patients (50%) were classified in the improving group (Group I) and 8 (50%) in the declining group (Group D). Patients from Group I and Group D, showing pulmonary function trajectories (Figure 1). The subjects were classified in the same groups based on annual changes in %FEV1 (Table 2). However, 3 subjects—#2, #14, and #16—were classified into the different groups by the changes in FEV1/FVC (data not shown).

### 3.2. Clinical Characteristics

The clinical characteristics, lung function, and laboratory data at the start of SCIT were compared between the two groups (Table 3). The mean age of the subjects was similar in Groups I and D, 10.5 and 11.8 years old, respectively. However, age at diagnosis of asthma was significantly younger in Group I than D. All 8 subjects in Group I were boys, whereas the subjects in Group D were 4 boys and 4 girls. Most of the subjects in both groups had co-morbid allergic rhinitis. The median fluticasone propionate-equivalent ICS dose at the start of SCIT was the same in the two groups, i.e., 200 µg/day and decreased at the last observation point. The FEV1% predicted, FEV1/FVC, MEF50% predicted, fractional exhaled nitric oxide (FeNO), total serum IgE level, and specific IgE levels for common inhaled allergens, except for HDM, were also similar in the two groups. The peripheral blood eosinophil count was significantly higher in Group D than in Group I. In contrast, the median HDM-specific IgE level (HDM-sIgE) was significantly higher in Group I than in Group D. Figure 2A plots the eosinophil counts and HDM-sIgE concentrations in Groups I and D and suggests independent association of the two factors with the treatment outcome.

Lung function at the last observation point were also compared (Table 3). Increase in mean values of lung function and decrease in median value of FeNO were observed in Group I; the opposite was true for Group D, although there were no statistical differences. 

### 3.3. Factors Associated with a Favorable Lung Function Trajectory with HDM-SCIT

Based on the above results, we constructed a logistic regression model for predicting a favorable lung function trajectory (Group I) using the eosinophil count and HDM-sIgE level at the start of HDM-SCIT (Table 4). The predicted probabilities for the subjects in the two groups are plotted in Figure 2B, and it is seen that the model clearly differentiates the lung function outcomes. The area under the ROC curve (Figure 2C) and the Hosmer–Lemeshow test also indicate high performance of the model (Table 4).

### 3.4. Clinical Outcome of the Subjects

Because SCIT is not recommended in patients with poorly controlled asthma, all the subjects were under good control at the start of HDM-SCIT. At the last observation point during the treatment period, asthma control remained good in all of the patients based on the international guideline [24]. However, one patient in Group D had occasional mild symptoms, which would be classified as partial control based on the more stringent Japanese guidelines [25]. The ICS dose was significantly lower at the last observation than at the start (Figure 3). The reduction in ICS appeared to be more prominent in Group D, but the difference was not statistically significant. No systemic adverse events related to HDM-SCIT were observed.

## 4. Discussion

The benefit of AIT for lung function in allergic asthma has not been established, possibly because of heterogeneous study patient populations. In this study, we sought to identify factors that relate to a beneficial effect of HDM-AIT on lung function. For evaluation of lung function, we focused on the longitudinal changes rather than measurement results at the last observation of AIT since lung function measurements at single time-points often fluctuate in children, and the long-term trend may better represent outcome. We found that a low peripheral blood eosinophil count and a high HDM-sIgE level at the start of the treatment were independently associated with an improved lung function trajectory in children and adolescents with HDM-allergic asthma. 

Why would a lower peripheral blood eosinophil count be associated with a favorable lung function trajectory? Eosinophils contribute to the pathogenesis of asthma by causing chronic airway inflammation [26], and a high peripheral blood eosinophil count is a characteristic feature of asthma [27]. A large epidemiological study found that a higher blood eosinophil count was associated with poor asthma outcomes, including asthma control [28]. A meta-analysis showed that a high blood eosinophil count was associated with higher odds of asthma exacerbation [29]. In addition, several studies reported that a high eosinophil count was associated with lung function decline [30,31,32], probably because eosinophils are involved in airway remodeling in asthma [26]. Although AIT has potential to control eosinophilic inflammation [33], it is positioned only as an adjunct therapy [7] and is not as potent as anti-eosinophil therapy using biologics [34]. Since continuous treatment with ICS was shown to control eosinophilic inflammation [35], we surmise from our current findings that a beneficial effect may be expected in the case of a low baseline blood eosinophil count. 

Higher levels of HDM-sIgE were also associated with a favorable HDM-SCIT outcome. For AIT, it is essential that the treatment be specific for the allergen that is causing the symptoms [36]. In this context, the subjects in this study were well-suited for HDM-SCIT because they had high sensitization levels to HDM and low sensitization levels to other common inhalant allergens (except for Japanese cedar pollen, which rarely causes asthma symptoms). Further, it was reported that a high sIgE/total IgE ratio predicted a better clinical response to AIT [37]. In agreement with that, the HDM-sIgE/total IgE ratio was significantly higher in Group I than in Group D (data not shown). Collectively, these findings may indicate that future studies of the effect of AIT on lung function may reach more definitive conclusions than past studies if the prospective subjects are stratified using these markers.

An additional important finding in this study was that the ICS dose was significantly lower (Figure 3) after about 4 years of HDM-SCIT than before the therapy, even while good asthma control was maintained. This agrees with previous systematic reviews that demonstrated that AIT significantly reduced medication use, especially in children [5,38].

In addition, it is noteworthy that the median dose of ICS was lower after HDM-SCIT not only in Group I but also in Group D (not statistically significant). It may be that if the higher dose were maintained [35] until the eosinophil count was adequately controlled, lung function decline might be prevented in the latter group. This possibility is worthy of further study.

This study has at least four limitations. First, the number of the subjects was too small. A much larger number is necessary to confirm the statistical difference observed in the study. However, we were able to observe the patients for changes for up to 7–8 years, and the findings from this long-term observation can be analyzed in a further RCT. Second, this was a retrospective observational study, without controls. Randomized placebo-controlled trials to exclude confounding factors that may impact the long-term effect of AIT need to be performed. Third, this study’s total observation period, although around 7 to 8 years, was not long enough. As stated above, the clinical impact of lung function decline becomes evident when children and adolescents reach older adulthood [10,11,12,13]. Fourth, classification of lung function changes by FEV1/FVC in 3 subjects did not match classification by MEF50% predicted. Classification by FEV1% predicted was well matched with that of MEF50% predicted. Since coefficients of variation (CV) for MEF50% predicted, FEV1% predicted, and FEV1/FVC were 28.4, 11.8, and 8.7, respectively, we assumed that MEF50% predicted (with the highest CV) better represented the lung function changes than FEV1/FVC (with the lowest CV).

## 5. Conclusions

This study identified two markers that seem to predict a beneficial effect of HDM-SCIT on lung function of children and adolescents with asthma. The predictive power of a lower peripheral blood eosinophil count may indicate the importance of control of airway inflammation for achieving the most benefit from the treatment. Similarly, the predictive power of a high HDM-sIgE level may underscore the importance of allergen specificity in considering AIT.

## Figures and Tables

**Figure 1 children-09-00487-f001:**
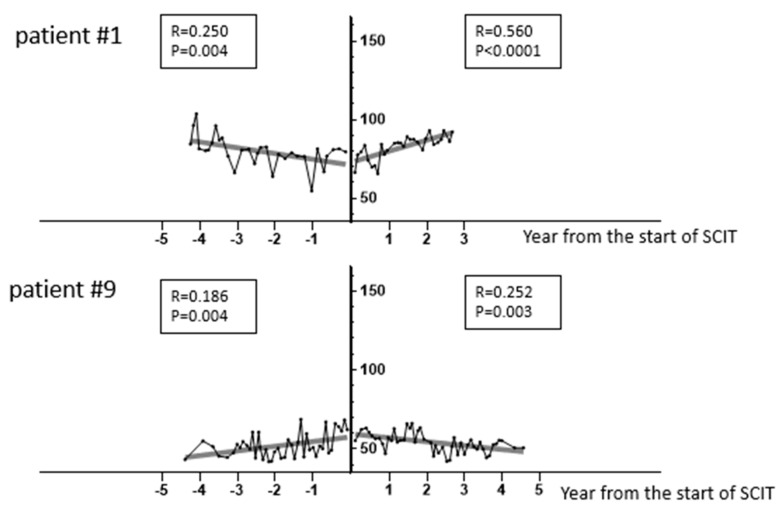
Representative lung function trajectories. MEF50% predicted values during the observation period are plotted (polygonal lines), and linear regression lines before and during SCIT are depicted in representative patients (#1 and #9 in Table 1, respectively) in Group I and Group D.

**Figure 2 children-09-00487-f002:**
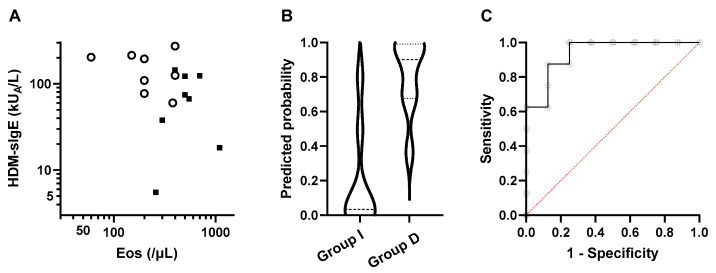
Peripheral blood eosinophil count (Eos) and HDM-sIgE as possible markers for predicting lung function outcomes. (**A**) The peripheral blood eosinophil counts (Eos) and HDM-sIgE levels in Group I (open circles) and Group D (closed squares) are plotted. (**B**) Violin plot graphs of the predicted probability calculated for each group by logistic regression analysis. (**C**) ROC curve for the logistic model.

**Figure 3 children-09-00487-f003:**
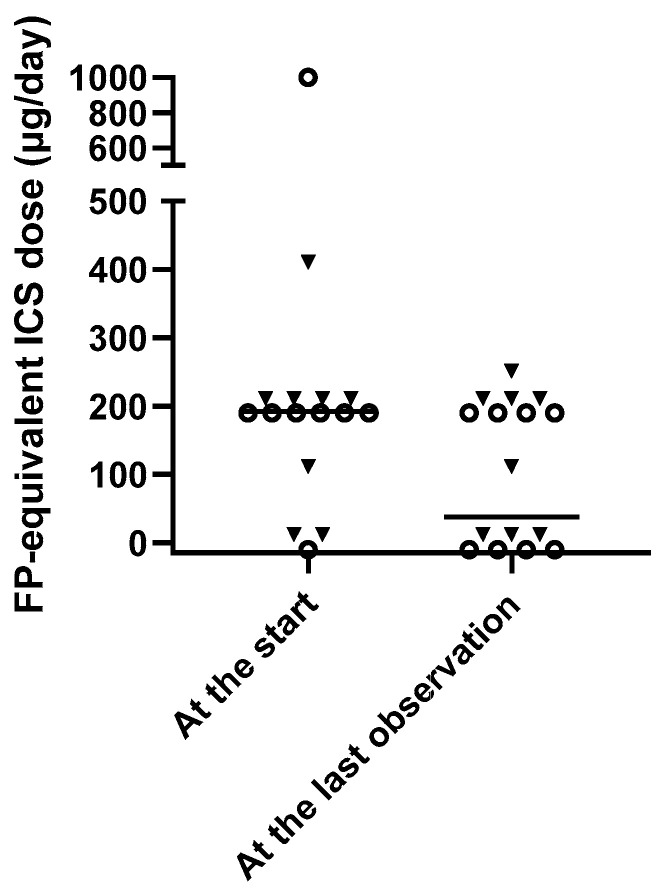
Fluticasone propionate (FP)-equivalent ICS doses at the start and at the last observation of HDM-SCIT (at the last visit of each patient in the study period). Open circles indicate subjects in Group I, while closed squares indicate subjects in Group D. In all subjects, the difference between the doses at the 2 time points was statistically significant. *p* < 0.05, Mann–Whitney U test.

**Table 1 children-09-00487-t001:** Predicted annual change in MEF50%.

ID	Group I		ID	Group D	
	Before SCIT	During SCIT		Before SCIT	During SCIT
1	−3.8	2.4	9	3.0	−1.8
2	1.3	1.4	10	6.7	−4.6
3	−9.0	1.1	11	4.4	−0.3
4	−0.8	−0.6	12	8.0	−15.2
5	−3.7	7.0	13	−3.7	−7.0
6	−11.3	−1.3	14	13.0	4.8
7	−10.9	−8.3	15	7.6	−0.4
8	−5.0	−2.5	16	10.6	3.1
Mean	−5.4	−0.1	Mean	6.2	−2.7

%/year.

**Table 2 children-09-00487-t002:** Predicted annual change in %FEV1.

ID	Group I		ID	Group D	
	Before SCIT	During SCIT		Before SCIT	During SCIT
1	−4.3	1.9	9	0.7	−2.8
2	−0.4	5.6	10	3.2	−2.6
3	−9.1	−0.8	11	−0.9	−12.2
4	0.4	0.5	12	−0.8	−6.1
5	−3.2	4.4	13	−2.6	−2.9
6	−15.3	0.7	14	3.8	−0.5
7	−16.2	−6.2	15	3.9	−5.7
8	−8.1	−2.5	16	10.6	1.8
Mean	−7.0	0.5	Mean	2.2	−3.9

%/year.

**Table 3 children-09-00487-t003:** Comparison of clinical characteristics between the two groups.

Characteristic	Group I (*n* = 8)	Group D (*n* = 8)	*p* Value
Age (years); mean ± SD			
at the start of SCIT	10.5 ± 2.0	11.8 ± 1.9	0.252
at diagnosis of asthma	6.4 ± 1.8	8.6 ± 1.8	0.033
Gender; M/F	8/0	4/4	0.077
Observation period (years); mean (range)			
Pre-SCIT	3.8 (0.6–7.5)	3.0 (1.6–7.2)	0.65
SCIT	4.1 (1.3–4.7)	3.9 (1.5–7.9)	0.74
Co-morbid allergic disease; *n* (%)			
Allergic rhinitis	8 (100%)	6 (75%)	0.467
Atopic dermatitis	2 (25%)	2 (25%)	>0.999
Food allergy	4 (50%)	2 (25%)	0.608
Pharmacological treatment at the start of SCIT			
Median dose of ICS (range)	200 (0–1000)	200 (0–400)	0.563
No use of ICS	1 (13%)	2 (25%)	>0.999
Use of omalizumab	2 (25%)	1 (13%)	>0.999
Pharmacological treatment at the last visit of SCIT			
Median dose (range)	100 (0–200)	150 (0–250)	0.563
No use of ICS	4 (50%)	3 (38%)	>0.999
Use of omalizumab	2 (25%)	1 (13%)	>0.999
Lung function (A) at the start and (B) at the last visit of SCIT			
FEV1% predicted (A)	85.8 ± 5.9	85.2 ± 8.1	0.958
(B)	88.6 ± 7.6	83.5 ± 8.6	0.156
FEV1/FVC ratio (%); mean ± SD (A)	90.8 ± 8.2	92.1 ± 8.9	0.713
(B)	92.8 ± 15.4	87.7 ± 10.5	0.637
MEF50% predicted; mean ± SD (A)	78.3 ± 13.7	94.7 ± 31.1	0.318
(B)	85.2 ± 23.6	85.5 ± 27.6	0.958
FeNO (ppb); median (range) (A)	33 (8–80)	25 (13–82)	>0.999
(B)	29 (16–105)	47 (6–107)	0.793
Blood eosinophil count at the start of SCIT (/µL); median (range)	200 (60–400)	500 (260–1100)	0.006
Total serum IgE at the start of SCIT (IU/mL); median (range)	652 (105–2976)	559 (135–2572)	0.959
Specific IgE (kUA/L) at the start of SCIT; median (range)			
HDM (*Dermatophagoides pteronyssinus*)	160 (60–273)	70 (5.5–144)	0.038
Japanese cedar pollen	13.1 (4.7–221)	6.7 (0.1–144)	0.328
Dog dander	0.3 (0.1–34.6)	0.4 (0.1–47.4)	0.485
Cat dander	0.1 (0.1–1.7)	0.3 (0.1–10.9)	0.114
Ragweed	0.2 (0.1–3.9)	0.3 (0.1–1.7)	0.657
Orchard grass	0.2 (0.1–29.3)	0.3 (0.1–8.1)	0.797

ICS dose: fluticasone propionate-equivalent dose (µg/day). FEV1/FVC ratio: forced expiratory volume at 1 s/forced expiratory volume ratio. MEF50 →MEF50%: maximal expiratory flow at 50% of FVC. FeNO: fractional exhaled nitric oxide. ppb: parts per billion.

**Table 4 children-09-00487-t004:** Logistic regression model to predict a favorable lung function trajectory.

**Parameter**	**Variable**	**Estimate**	**95% CI**	**Odds Ratio**	**95% CI**
β1	Eos	0.017	0.003 to 0.053	1.017	1.003 to 1.054
β2	HDM IgE	−0.028	−0.078 to −0.002	0.973	0.925 to 0.998
			**Statistic**	**95% CI**	***p* value**
Area under the ROC curve			0.938	0.822 to 1.000	0.003
Hosmer–Lemeshow test			5.506		0.702

## Data Availability

The data are in computers that are not connected to the Internet. Whenever disclosure is needed, the authors are ready to provide the information.

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
