# Peer review of "House Dust Mite Subcutaneous Immunotherapy and Lung Function Trajectory in Children and Adolescents with Asthma"

_children, 2022, doi:10.3390/children9040487_

Round 1
Reviewer 1 Report
The paper „House dust mite subcutaneous immunotherapy and lung function trajectory in children and adolescents” by Nogami et al. aims to describe the influence of subcutaneous immunotherapy with allergens on ventilator parameters in young asthmatic patients. The study was performed on a small group of 16 subjects divided into an improving (I) and a non-improving (D) group. The authors conclude that a higher level of IgE against the allergen at the beginning of the therapy leads to an improvement of lung function trajectory.
There are several issues which I would like to address to the authors.
First of all, data on lung function (FEV1 pred, FEV1/FVC, FeNO etc) after AIT (allergen-specific immunotherapy) are missing in the paper and thus, it is very hard to prove whether AIT improved lung function (in group I) or not (group D). Has “lung function trajectory” (the term has not been explained in a clear way by the authors, please explain) been only estimated using pre-AIT parameters and not confirmed by lung function parameters at the last observation? If it was the case, how could the subjects be divided into the two groups?
If these data from the last observation are available, were they significantly different from the parameters at the beginning of AIT and did they vary between the groups? What did improve and what did worsen?
The authors included into their research subjects under 18 years of age. What was the range of age? At what age did AIT start? What was the age of diagnosis of asthma? Were there any differences in any of the parameters between pre-pubertal children and post-pubertal adolescents? Could the hormonal development of the subjects interfere/influence somehow the outcomes of AIT?
The authors point that the use of ICS decreased in the I but not in the D group. What about airway inflammation? Has it been measured? What could be the reason for a diminished ICS use? Could it have something to do with the increasing age of the patients?
Did the subjects have other allergies than to HDM? If yes, did they receive special treatment which could influence the main one?
Is AIT a good add-on therapy for all kind of allergic asthma and allergies of any type? Are there any serious risk factors? What are the mechanisms of its beneficial action? Maybe it would be an idea to include some more information into the introduction or discussion.
The authors conclude that subjects with higher IgE against HDM at the beginning of AIT had better outcomes. Could the authors comment on this result? Could AIT be a kind of booster for the immune system?
To sum up, data from the last observation (end of AIT) should be presented and discussed.
Author Response
Responses to reviewers’ comments.
Manuscript ID: children-1619342
We have revised the manuscript in consideration of the comments of Reviewer 1. Below, we present our responses to those comments.
The paper “House dust mite subcutaneous immunotherapy and lung function trajectory in children and adolescents” by Nogami et al. aims to describe the influence of subcutaneous immunotherapy with allergens on ventilator parameters in young asthmatic patients. The study was performed on a small group of 16 subjects divided into an improving (I) and a non-improving (D) group. The authors conclude that a higher level of IgE against the allergen at the beginning of the therapy leads to an improvement of lung function trajectory.
There are several issues that I would like to address to the authors.
Point 1: First of all, data on lung function (FEV1 pred, FEV1/FVC, FeNO etc) after AIT (allergen-specific immunotherapy) are missing in the paper and thus, it is very hard to prove whether AIT improved lung function (in group I) or not (group D).
Response 1: We have added the lung function data (FEV1 pred, FEV1/FVC, FeNO etc) after AIT (SCIT) in Table 2. Although there were no statistical differences, mean/median values of the spirometry data improved in group I, not in group D, and FeNO decreased in group I and increased in group D. Classification of trajectories by FEV1 % predicted matched that by MEF50 % predicted, indicating some validity of the classification criteria. However, classification by FEV1/FVC did not match that by the other 2 indices in 3 subjects. Since the coefficient of variation (CV) for MEF50 % predicted was highest among the 3 indices, we assume that it well represented the changes in lung function. In contrast, CV for FEV1/FVC was the lowest, which may indicate less reflectivity of the index. We have stated these facts as a limitation.
Point 2: Has “lung function trajectory” (the term has not been explained in a clear way by the authors, please explain) been only estimated using pre-AIT parameters and not confirmed by lung function parameters at the last observation? If it was the case, how could the subjects be divided into the two groups?
Response 2: We evaluated the linear trend of the changes in lung function before and after SCIT, then based on the slope value calculated for each subject, we divided them into 2 groups. The definition has been described from lines 94 to 100. Regarding the lung function at the last observation, we have added the data in Table 2. There were no differences in the lung function values. However, since we focused on longitudinal changes rather than values at a single time point, we believe that our analysis has a certain significance.
Point 3: If these data from the last observation are available, were they significantly different from the parameters at the beginning of AIT and did they vary between the groups? What did improve and what did worsen?
Response 3: As noted in the above section, there were no statistical differences in lung function data at the start and at the last visit of SCIT. However, because measures of lung function often fluctuate nonspecifically in children, we sought to identify trends in changes by performing linear regression analyses of the measured data over a long period of time. We believe that this approach has provided us with certain insights.
.Point 4: The authors included into their research subjects under 18 years of age. What was the range of age? At what age did AIT start? What was the age of diagnosis of asthma? Were there any differences in any of the parameters between pre-pubertal children and post-pubertal adolescents? Could the hormonal development of the subjects interfere/influence somehow the outcomes of AIT?
Response 4: The range of age in this study was childhood and adolescence, when lung function is growing. The age at the start of SCIT has been described in Table 2. In addition, we have added the age at diagnosis of asthma. Interestingly, the age at the diagnosis in Group I was significantly younger than Group D. Usually, in childhood asthma, the inception of recurrent wheezing is sometime during the preschool period and the definitive diagnosis is made when a child reaches school age. Since controller treatment starts at the diagnosis of asthma, younger age of diagnosis may implicate earlier treatment. which may have benefited the patients resulting in a favorable lung function trajectory. However, the above notion is quite speculative and was not included in the text.
We were not able to analyze the hormonal effect on asthma outcome because clinical records on pubertal changes in the subjects were not obtained.
Point 5: The authors point that the use of ICS decreased in the I but not in the D group. What about airway inflammation? Has it been measured? What could be the reason for a diminished ICS use? Could it have something to do with the increasing age of the patients?
Response 5: ICS dose was adjusted according to the level of control as recommended by the guidelines and decreased in both groups during SCIT. We have added the ICS use at the last visit of SCIT in Table 2. The good control resulting in ICS dose reduction may be due to the treatment, or it may be just due to age. We also measured FENO as a marker of airway inflammation, but the guidelines do not recommend using FeNO as a criterion for ICS dose adjustment. What this study did show is that among patients who had their ICS dose reduced on the basis of good control, some had a “silent” decline in lung function, and we were able to identify a risk factor for this.
Point 6: Did the subjects have other allergies than to HDM? If yes, did they receive special treatment which could influence the main one?
Response 6: As shown in Table 2, most of the subjects had allergic rhinitis. We treated co-morbid allergic rhinitis with antihistamines and intranasal steroids according to their symptoms. Unfortunately, however, we did not analyze the effect of the treatment because the majority had similar medications.
Point 7: Is AIT a good add-on therapy for all kind of allergic asthma and allergies of any type? Are there any serious risk factors? What are the mechanisms of its beneficial action? Maybe it would be an idea to include some more information into the introduction or discussion.
Response 7: US asthma management guidelines recommend SCIT as an adjunct therapy to standard pharmacotherapy for mild to moderate allergic persistent asthma. Anaphylaxis is a rare but serious adverse event, which can be managed at well-equipped institutions. Major mechanisms for its efficacy are considered to be the induction of suppressive cytokines that leads to inhibition of inflammatory cascade and unresponsiveness to the allergen. We have added the above information in Introduction.
Point 8: The authors conclude that subjects with higher IgE against HDM at the beginning of AIT had better outcomes. Could the authors comment on this result? Could AIT be a kind of booster for the immune system?
Response 8: We discussed the possible implication of high HDM-specific IgE for the beneficial outcome of SCIT in the third paragraph in Discussion. We believe that higher specific IgE represents higher specificity to HDM. Since allergen-specificity is the requisite for AIT, we believe that higher specificity leads to higher efficacy. It has been reported that specific IgE increases after initiation of AIT, later it decreases as AIT as a result of AIT also suppressing antibody production.
Point 9: To sum up, data from the last observation (end of AIT) should be presented and discussed.
Response 9: As noted above, we have added the data at the last observation and summarized them in the Results and Discussion sections.
“For evaluation of lung function, we focused on the longitudinal changes rather than measurement results at the last observation of AIT since lung function measurements at single time-points often fluctuate in children and long-term trend may better represent outcome.”

Reviewer 2 Report
The paper "House dust mite subcutaneous immunotherapy and lung function trajectory in children and adolescents with asthma" by Kazutaka Nogami et all provides results from a long follow-up of a relatively small group of children and adolescents with asthma and lung function measurements. The major plus in a duration of the observation. There are issues that can be solved.
- Page 2, reference 18 needs an update: Busse WW, Szefler SJ, Haselkorn T, Iqbal A, Ortiz B, Lanier BQ, Chipps BE. Possible Protective Effect of Omalizumab on Lung Function Decline in Patients Experiencing Asthma Exacerbations. J Allergy Clin Immunol Pract. 2021 Mar;9(3):1201-1211. doi: 10.1016/j.jaip.2020.10.027. Epub 2020 Oct 24. PMID: 33223095.
- Page 2, Lung function. Do you have data about the quality of measurements? MEF 50 is an accurate value as recommended but the technical issues are important in children. ERS/ATS spirometry criteria? 3 best measurements out of max 8?
- Page 3, results. 16 patients with 1158 valid spirometric tests. Can you define the "valid" term?
- Page 6, reference 22. Can be edited or completed: Singh D, Agusti A, Anzueto A, Barnes PJ, Bourbeau J, Celli BR, Criner GJ, Frith P, Halpin DMG, Han M, López Varela MV, Martinez F, Montes de Oca M, Papi A, Pavord ID, Roche N, Sin DD, Stockley R, Vestbo J, Wedzicha JA, Vogelmeier C. Global Strategy for the Diagnosis, Management, and Prevention of Chronic Obstructive Lung Disease: the GOLD science committee report 2019. Eur Respir J. 2019 May 18;53(5):1900164. doi: 10.1183/13993003.00164-2019. PMID: 30846476.
- Page 7, 3.5. Is the same as 3.4, page 6.
- Page 8. There are important limitations but you can add that despite a limited number of patients observed for up to 7-8 years, there are tendencies that can be analysed in further RCT.
- Page 6, figure 3. It is not clear when was the last observation.
- Page 9, reference 18 can be updated. Busse WW, Szefler SJ, Haselkorn T, Iqbal A, Ortiz B, Lanier BQ, Chipps BE. Possible Protective Effect of Omalizumab on Lung Function Decline in Patients Experiencing Asthma Exacerbations. J Allergy Clin Immunol Pract. 2021 Mar;9(3):1201-1211. doi: 10.1016/j.jaip.2020.10.027. Epub 2020 Oct 24. PMID: 33223095.
- Page 9, reference 33. Needs update: Vähätalo I, Ilmarinen P, Tuomisto LE, Tommola M, Niemelä O, Lehtimäki L, Nieminen P, Kankaanranta H. 12-year adherence to inhaled corticosteroids in adult-onset asthma. ERJ Open Res. 2020 Mar 23;6(1):00324-2019. doi: 10.1183/23120541.00324-2019. PMID: 32211439; PMCID: PMC7086072.
Copy
Author Response
Responses to reviewers’ comments.
Manuscript ID: children-1619342
We have revised the manuscript in consideration of the comments of Reviewer 1. Below, we present our responses to those comments.
Comments and Suggestions for Authors
The paper "House dust mite subcutaneous immunotherapy and lung function trajectory in children and adolescents with asthma" by Kazutaka Nogami et all provides results from a long follow-up of a relatively small group of children and adolescents with asthma and lung function measurements. The major plus in the duration of the observation. There are issues that can be solved.
Point 1: Page 2, reference 18 needs an update: Busse WW, Szefler SJ, Haselkorn T, Iqbal A, Ortiz B, Lanier BQ, Chipps BE. Possible Protective Effect of Omalizumab on Lung Function Decline in Patients Experiencing Asthma Exacerbations. J Allergy Clin Immunol Pract. 2021 Mar;9(3):1201-1211. doi: 10.1016/j.jaip.2020.10.027. Epub 2020 Oct 24. PMID: 33223095.
Response 1: We have updated the reference.
Point 2: Page 2, Lung function. Do you have data about the quality of measurements? MEF 50 is an accurate value as recommended but the technical issues are important in children. ERS/ATS spirometry criteria? 3 best measurements out of max 8?
Response 2: We performed spirometry in compliance with the technical statement from ERS/ATS (Eur Respir J 2005; 26: 319-38). Accordingly, we adopted the best data out of a minimum of 3 maneuvers (no more than 8).
Point 3: Page 3, results. 16 patients with 1158 valid spirometric tests. Can you define the "valid" term?
Response 3: We identified outliers by the ROUT method (Motulsky, BMC Bioinformatics 2006; 7: 123.) and excluded 7 data out of the total 1165.
We revised the sentence as follows:
“A total of 16 patients were eligible for the study, and we analyzed 1,158 spirometric measurements after excluding 7 outliers out of the total 1165 by the ROUT method with a false discovery rate <1%, [38].”
Point 4: Page 6, reference 22. Can be edited or completed: Singh D, Agusti A, Anzueto A, Barnes PJ, Bourbeau J, Celli BR, Criner GJ, Frith P, Halpin DMG, Han M, López Varela MV, Martinez F, Montes de Oca M, Papi A, Pavord ID, Roche N, Sin DD, Stockley R, Vestbo J, Wedzicha JA, Vogelmeier C. Global Strategy for the Diagnosis, Management, and Prevention of Chronic Obstructive Lung Disease: the GOLD science committee report 2019. Eur Respir J. 2019 May 18;53(5):1900164. doi: 10.1183/13993003.00164-2019. PMID: 30846476.
Response 4: We have updated the reference.
Point 5: Page 7, 3.5. Is the same as 3.4, page 6.
Response 5: Thank you for pointing this out. We have corrected it.
Point 6: Page 8. There are important limitations but you can add that despite a limited number of patients observed for up to 7-8 years, there are tendencies that can be analyzed in further RCT.
Response 6: We appreciate this encouraging comment. We have added the following sentence:
“However, we were able to observe the changes for up to 7-8 years, and the findings from this long-term observation can be analyzed in further RCT.”
Point 7: Page 6, figure 3. It is not clear when was the last observation.
Response 7: The last observation was the last visit of each patient in this study period.
Point 8: Page 9, reference 18 can be updated. Busse WW, Szefler SJ, Haselkorn T, Iqbal A, Ortiz B, Lanier BQ, Chipps BE. Possible Protective Effect of Omalizumab on Lung Function Decline in Patients Experiencing Asthma Exacerbations. J Allergy Clin Immunol Pract. 2021 Mar;9(3):1201-1211. doi: 10.1016/j.jaip.2020.10.027. Epub 2020 Oct 24. PMID: 33223095.
Response 8: We have updated the reference.
Point 9: Page 9, reference 33. Needs update: Vähätalo I, Ilmarinen P, Tuomisto LE, Tommola M, Niemelä O, Lehtimäki L, Nieminen P, Kankaanranta H. 12-year adherence to inhaled corticosteroids in adult-onset asthma. ERJ Open Res. 2020 Mar 23;6(1):00324-2019. doi: 10.1183/23120541.00324-2019. PMID: 32211439; PMCID: PMC7086072.
Response 9: We have updated the reference.

Reviewer 3 Report
Authors wished to investigate the characteristics of patients (children and adolescents) who showed improvement in MEF50 during AIT-HD compared to patients who did not show any improvement in MEF50. They found that patients who improved had a significantly lower blood eosinophil count and a significantly higher HDM-specific IgE level compared to patients who did not improve, with no difference in control of asthma and in asthma drug use.
Unfortunately the retrospective design of the study, the very small number of enrolled patients (8 responders and 8 non responders!), the arbitrary choice of the timing of spirometric data (particularly before AIT) make the study scientifically unsound.
Critical points
- Authors should explain how they select the patients to be enrolled into the study and why they enrolled only 16 patients.
- The time points where spirometry had been obtained should be explained. How asthma therapy and exacerbations (at time points of spirometry) had been taken into account should be explained in details.
- As asthma was controlled in both groups, who used less inhaled ICS during AIT and showed normal MEF50 values, the clinical meaning of change in MEF50 over time is not clear. The patients of both groups reached the same MEF50 values during AIT!
Author Response
Responses to reviewers’ comments.
Manuscript ID: children-1619342
We have revised the manuscript in consideration of the comments of Reviewer 2. Below, we present our responses to those comments.
Comments and Suggestions for Authors
Authors wished to investigate the characteristics of patients (children and adolescents) who showed improvement in MEF50 during AIT-HD compared to patients who did not show any improvement in MEF50. They found that patients who improved had a significantly lower blood eosinophil count and a significantly higher HDM-specific IgE level compared to patients who did not improve, with no difference in control of asthma and in asthma drug use.
Point 1: Unfortunately, the retrospective design of the study, the very small number of enrolled patients (8 responders and 8 nonresponders!), the arbitrary choice of the timing of spirometric data (particularly before AIT) make the study scientifically unsound.
Response 1: We appreciate this comment. We are fully aware that the number of patients is very small, but we believe that long-term observation of up to 7 years has a certain significance. In addition, we enrolled only patients whose lung function had been observed for at least 6 months prior to SCIT and at least 1 year after SCIT, and we adopted all spirometry data that were measured according to ERS/ATS standards. No arbitrary selection of data was performed.
Critical points
Point 2: Authors should explain how they select the patients to be enrolled into the study and why they enrolled only 16 patients.
Response 2: As stated above, we enrolled all the patients whose lung function had been observed for at least 6 months before SCIT and at least 1 year after SCIT, and we adopted all spirometry data that were measured according to the ERS/ATS standards.
Since SCIT is considered to be more effective than SLIT, we, as a specialized hospital, have been actively providing SCIT. Unfortunately, however, patients tended to prefer SLIT to SCIT because of a painful nature of SCIT, and only 16 patients were able to be observed for a long period of time.
Point 3: The time points where spirometry had been obtained should be explained. How asthma therapy and exacerbations (at time points of spirometry) had been taken into account should be explained in detail.
Response 3: Since SCIT should be performed when asthma is under control, all the measurements were performed when the patients were asymptomatic and stable. We added this to Methods.
Point 4: As asthma was controlled in both groups, who used less inhaled ICS during AIT and showed normal MEF50 values, the clinical meaning of change in MEF50 over time is not clear. The patients of both groups reached the same MEF50 values during AIT!
Response 4: In daily clinical practice, asthma control and respiratory function are assessed at each visit, and treatment is adjusted accordingly. Usually, it is not easy to evaluate long-term changes in lung function, and it is often not performed. However, recent long-term cohort studies have raised concerns that poor growth of lung function during childhood may predispose to low lung function in adulthood, increasing the risk for COPD. This was the impetus for conducting the present study. In consideration of the possible long-term consequences, we believe that even small changes in childhood can have significance later in life.
Round 2
Reviewer 1 Report
I do not have any further comments.
Reviewer 3 Report
Unfortunately also the revised manuscript cannot change the limitations of the study.
In particular, the number of enrolled patients is too small ! Spirometry had been obtained always during asthma control, but no mention of exacerbation has been provided, as well non mention of therapy at time of spirometry.
Authors observed different change in MEF 50, but the final values were similar (and normal !) in both groups and no mention of FEV-1 change has been provided.
Author Response
Responses to reviewers’ comments.(Round 2)
Manuscript ID: children-1619342
We have revised the manuscript in consideration of the comments of Reviewer 1. Below, we present our responses to those comments.
Point 1: Unfortunately also the revised manuscript cannot change the limitations of the study.
In particular, the number of enrolled patients is too small ! Spirometry had been obtained always during asthma control, but no mention of exacerbation has been provided, as well non mention of therapy at time of spirometry.
Response 1: None of the subjects experienced asthma exacerbation during HDM-SCIT (described at lines 179-183), thus no spirometer results at exacerbation. We have added information on controller medications at the start and at the last visit of HDM-SCIT. We recorded pharmacotherapy at each spirometry measurement. However, since all the subjects were under control, there were no frequent changes in medication except for a gradual step down of treatment, which, we believe, can be summarized as medication at the last visit.
It is true that the number of patients was small, but we still obtained some statistically significant results, so we thought that even a summary with a small number of subjects would be useful for future large-scale trials.
Point 2: Authors observed different change in MEF 50, but the final values were similar (and normal !) in both groups and no mention of FEV-1 change has been provided.
Response 2: We have added the lung function data (FEV1 pred, FEV1/FVC, FeNO etc) after AIT (SCIT) in Table 2. Although there were no statistical differences, mean/median values of the spirometry data improved in group I, not in group D and FeNO decreased in group I and increased in group D. Classification of trajectories by FEV1 % predicted matched that by MEF50 % predicted, indicating some validity of the classification criteria. However, classification by FEV1/FVC did not match that by the other 2 indices in 3 subjects. Since the coefficient of variation (CV) for MEF50 % predicted was highest among the 3 indices, we assume that it well represented the changes in lung function. In contrast, CV for FEV1/FVC was the lowest, which may indicate less reflectivity of the index. We have stated these facts as a limitation.
There were no statistical differences in lung function data at the start and at the last visit of SCIT. However, because measures of lung function often fluctuate nonspecifically in children, we sought to identify trends in changes by performing linear regression analyses of the measured data over a long period of time. We believe that this approach has provided us with certain insights.
Round 3
Reviewer 3 Report
Authors recognize that there were no statistical differences in lung function data at the start and at the last visit of SCIT and this result is not changed by the analysis of trends in functional changes, a type of analysis never used before!